# Unwinding on the Weekend from Work-Related Stress: Moderating Effect of Weekday Psychological Stress on the Relationship between Increased Recovery Experience and Reduction of Psychological Stress on the Weekend

**DOI:** 10.3390/bs12060163

**Published:** 2022-05-25

**Authors:** Satoshi Horiuchi, Suguru Iwano, Shuntaro Aoki, Yuji Sakano

**Affiliations:** 1Department of Social and Clinical Psychology, Hijiyama University, Hiroshima 732-8509, Japan; 2Faculty of Welfare and Health Science, Oita University, Oita 870-1192, Japan; iwano-suguru@oita-u.ac.jp; 3Center for Medical Education and Career Development, Fukushima Medical University, Fukushima 960-1295, Japan; aokishuntaro1292@gmail.com; 4Department of Neuropsychiatry, Fukushima Medical University, Fukushima 960-1295, Japan; 5Department of Clinical Psychology, Goryokai Medical Corporation, Sapporo 002-8029, Japan; sakano3yuji@gmail.com; 6Sapporo CBT & EAP Center, Sapporo 002-8028, Japan

**Keywords:** psychological stress response, recovery, relaxation, school teachers, weekend

## Abstract

Recovery is described as a process that is in contrast to the process whereby the psychological stress response increases. Recovery experiences refer to specific experiences that promote recovery and represent psychological attributes including relaxation. This preliminary study tested the hypothesis that levels of psychological stress before the weekend have a moderating effect on the relationship between an increased recovery experience during the weekend and a reduction of psychological stress from workdays to the weekend. Of 270 Japanese teachers who were invited to participate, 181 completed questionnaires on the psychological stress response before, on the psychological stress response and the recovery experience during the weekend. Data from 7 part-time teachers and 38 teachers who were not stressed at all before the weekend were excluded; therefore, data for 136 participants were ultimately analyzed. Results of hierarchal regression analysis indicated that increased relaxation was associated with an increased reduction in psychological stress response during the weekends in participants with high levels of psychological stress before the weekends. This moderating effect was not observed for other recovery experiences. Considering the psychological stress response is important for research on recovery experiences during weekends.

## 1. Introduction

Previous research has suggested that weekends can serve as an opportunity for recovery from work-related stress [1,2,3], where recovery is described as a process in contrast to strain [4]. Previous studies have reported that the intensity of negative effects decreases from workdays to weekends [5,6,7], whereas respite from work during weekends can be an inexpensive and accessible means of managing stress [8]. An increased understanding of decreases in the level of stress response from workdays to weekends, or the occurrence of recovery during weekends, can shed light on the means by which people spend weekends to manage psychological stress response effectively. More specifically, psychological stress response refers to a variety of emotional (affective), behavioral, and cognitive responses induced by stressors [9].

Moreover, previous studies proposed that recovery experience [10] is one of the mechanisms through which recovery from stress occurs [8,11,12,13]. It refers to specific experiences that promote recovery and represent psychological attributes. In a study by Sonnentag and Fritz [10], four recovery experiences were proposed, namely, psychological detachment (experience of being not only mentally but also physically away from work), relaxation (experience of being relaxed), control (experience of choosing what to do), and mastery (experience of learning something new or challenging). Previous studies have reported that, out of the four experiences, frequent relaxation during weekends has been consistently associated with better mental health status during weekends [12] and after weekends [8]. In this vein, Fritz et al. [12] have examined the associations between recovery experience during weekends and the four types of negative affect, namely, hostility, fear, sadness, and fatigue, within the same period in German preschool teachers. The results showed that only relaxation on weekends out of the four recovery experiences was associated with low levels of hostility and fear, whereas none of the recovery experiences were related to either fatigue or sadness.

However, less is known about the factors that moderate the relationship between recovery experience and reduction of the psychological stress response. Some longitudinal studies have focused on the relationships between recovery experiences during the weekend and the psychological stress response. For example, Drach-Zahavy and Marzuq [14] examined the moderating effects of the timing of respite (i.e., on the weekend or midweek days) on the relationship between recovery experiences and emotional exhaustion and vigor after respite among nurses. Ragsdale et al. [8] evaluated the integrated model of stress and recovery during a weekend in a sample of undergraduates. This model assumed two variables that mediated, not moderated, the relationships between recovery experiences during the weekend and psychological strain after the weekend. Similarly, Cho and Park [11] examined weekend recovery experiences and the negative effects after the weekend, along with physical activity. Cho and Park [11] focused on the effects of the weekend physical activity on the negative effects after the weekend, and employed recovery experiences as the mediator of the effect. Fritz et al. [12] and Hahn et al. [15] examined the relationships between recovery experiences during the weekend and affective status during and after the weekend; however, these studies did not examine the factors that moderate these relationships.

One possible but less researched factor is the level of psychological stress before the weekend. In other words, weekend recovery experience could be more strongly related to a reduction of psychological stress response during the weekend in individuals with higher stress levels, in contrast to low pre-weekend levels of psychological stress response. Siltaloppi et al. [16] pointed to several factors that have moderating effects on the relationship between daily recovery experience and reduction of the psychological stress response. The authors reported that psychological detachment and mastery were strongly associated with an increased need for recovery in workers with low levels of job control. In addition, relaxation was strongly related to increased job exhaustion in workers with a high time demand. Low levels of job control and high time demand are stressors that have been associated with increases in psychological stress response [17]; therefore, such studies have hypothesized that levels of psychological stress before the weekend will have moderating effects on the relationships between increased psychological detachment, relaxation, and mastery during the weekend and reduced psychological stress between workdays and the weekend. However, studies that directly tested this hypothesis remain scarce.

Testing this hypothesis is potentially important as its results could demonstrate the value of considering the extent of the stress of participants in the research on recovery experience. The majority of previous studies that examined recovery experience during weekends [5,8,9,10] might have overlooked the strength of the psychological stress response and analyzed data from all participants including those who were not stressed. Recovery has been identified as a process that stands in opposition to strain; therefore, psychological stress levels do not decrease, and recovery during weekends would only occur if workers were stressed before the weekend. Seemingly, the relationship between recovery experience and reduction in psychological stress response is demonstrated more clearly if researchers assess psychological stress levels and exclude individuals who are not determined to be highly stressed; therefore, studies that focus on recovery experience during weekends should consider the extent of the psychological stress response in participants and exclude those with low levels of stress.

This study is preliminary in nature. It examines whether levels of psychological stress before the weekend had a moderating effect on the relationship between increased recovery experience and reduced psychological stress between workdays and the weekend in a sample of Japanese teachers. The sample represents one of the populations in which managing occupational stress is important [18]. The results of the 2013 Teaching and Learning International Survey conducted by the Organization for Economic Cooperation and Development [19] demonstrated that Japanese teachers in secondary elementary schools worked for 53.9 h per week. This duration of working hours was one of the longest among the 34 participating countries and regions, as the legal working hours for public school teachers were 38.75 h per week. Long working hours have been reported to be associated with increased risks of mental health problems, such as depressive states and anxiety [20]. Based on the results of studies conducted by Fritz et al. [12] and Siltaloppi et al. [16], this present study hypothesized that increased psychological detachment, relaxation, and mastery will be associated with an increased reduction of the psychological stress response during the weekend when levels of psychological stress before the weekend are noted to be high.

## 2. Materials and Methods

### 2.1. Participants

The participants were recruited by research volunteers who were schoolteachers or family members or friends of the participants. In total, 270 schoolteachers were invited to participate. Thorough written and verbal explanations regarding the procedure, the objective of the study was provided. In addition, the following information regarding their rights was disclosed: (a) the survey is completely voluntary, and (b) they may refuse to participate in the survey or withdraw consent even after agreeing to participate. Out of the 270 schoolteachers, 213 completed the questionnaire (11 pages), whereas 57 declined. Furthermore, out of these 213 teachers, 181 provided complete data, whereas data for seven part-time teachers were excluded because they represented a very small proportion of the participants. Thus, the final analysis consisted of data from 174 participants.

### 2.2. Measures

The questionnaire is composed of the items and scales discussed in the following sections. The participants completed the questionnaire after work on a Friday (Time 1 (T1)) and before going to bed the following Sunday (Time 2 (T2)). The order of the presentation of the scales included in the questionnaire completed on Sunday was not counterbalanced.

#### 2.2.1. Demographic Characteristics and Occupational Variables

Data were measured at T1, which included age, sex, type of school in which the participant worked, marital status, parenthood, type of teacher (i.e., nursing teacher or other types of teacher designation and managerial or non-managerial position), and average weekly overtime (in hours).

#### 2.2.2. Psychological Stress Response

The participants completed the Stress Response Scale-18 (SRS-18) [9] at T1 and T2. The SRS-18 consists of 18 items that are further divided equally between three subscales, namely, depression–anxiety, irritability–anger, and helplessness. The items were then rated using a four-point Likert-type scale ranging from 0 (no stress at all) to 3 (high stress levels) to measure feelings and behaviors experienced over a few days. At T2, the participants completed the SRS-18 items to reflect their feelings and behavior during the weekend (Friday evening to Sunday night). The scores for all items were totaled to provide the total score, wherein higher scores indicate higher levels of psychological stress. The SRS-18 has demonstrated reliability and validity with α ranging from 0.83 to 0.91 [9]. SRS-18 scores are standardized and can be converted to *T* scores (*M* = 50, *SD* = 10) according to sex to determine the scores of participants by sex; however, only raw scores were analyzed due to the relatively small sample size. For interpretation, the average scores for adult males and females were 14 and 16, respectively [9]. Cronbach’s alpha coefficient for the present sample was 0.95.

#### 2.2.3. Recovery Experience

Data were measured at T2 using the Japanese version of the REQ [4]. The scale consists of 16 items divided under four subscales, namely, psychological detachment, relaxation, control, and mastery. An example of an item from psychological detachment is “I forget about work”. One from relaxation is “I use the time to relax”. One from control is “I decide my own schedule”. One from mastery is “I learn new things”. Items were then rated using a 5-point Likert-type scale ranging from 1 (strongly disagree) to 5 (strongly agree) to best represent what each respondent did during the weekend. Scores for each subscale are totaled to provide the total score. The REQ-J has demonstrated reliability and validity [4]. Cronbach’s alpha coefficients for the present sample were 0.81, 0.87, 0.88, and 0.85 for psychological detachment, relaxation, control, and mastery, respectively.

#### 2.2.4. Life Stressors

Data were measured at T2 using Tanaka and Takagi’s [21] scale, which consists of four items and measures exposure to a broad range of stressors that occur in one’s personal life, such as caregiving burdens involving family members, housekeeping, and concerns about one’s health and family issues. Items were rated using a four-point Likert scale ranging from 1 (strongly disagree) to 4 (strongly agree). In the study, Cronbach’s α for the scale has been found to reach 0.65, which was considered acceptable.

#### 2.2.5. Weekend Work

The participants indicated whether they worked over the weekend. The response options for this question were “yes” or “no”. No information regarding the reliability or validity of the item was available; however, similar one-item questions have been used in previous studies that examined recovery during weekends [14,15].

### 2.3. Procedure

The institutional review board at the institution of the first author approved this study. The survey was conducted between October 2013 and March 2014 at elementary, junior high, high, and special-needs schools in Hokkaido (northern island), Tokyo Metropolis, Saitama prefecture (a prefecture in Kanto region in the island of Honshu), and Oita prefecture (a prefecture in Kyushu region, a southern island). After obtaining permission from school principals, the questionnaires were distributed to teachers from ten schools, where eight teacher volunteers and two graduate student volunteers managed the distribution. These eight teachers were also study participants. Written informed consent was not obtained from participants to maintain anonymity. Instead, the participants provided verbal informed consent. Furthermore, the researchers explained that submitting a completed questionnaire is equivalent to consent.

### 2.4. Statistical Analysis

Analyses were performed using SPSS for Windows version 24. As a preliminary analysis, the study identified participants with SRS-18 scores of 0, and their data were excluded from analysis. These participants reported that they were not stressed; therefore, recovery from stress could not occur from T1 to T2. The demographic and occupational characteristics of participants whose data were included in and excluded from the analysis were recorded and examined (Table 1). The means and standard deviations of the study variables and correlations between variables were also calculated (Table 2). A *t*-test was performed to determine whether levels of psychological stress at T2 were lower compared with those observed at T1. In addition, hierarchal regression analysis was conducted to verify the hypothesis (Table 3). The dependent variable was the change in the psychological stress response from Time 1 to Time 2, which was calculated by subtracting the Time 2 score from the Time 1 score. Greater values represent greater stress reduction. In Step 1, seven control variables (i.e., life stressors, weekend work, psychological stress level at T1, and four recovery experiences) were inputted as independent variables. In Step 2, interactions between psychological stress levels at T1 and the four recovery experiences were inputted. The centered product terms for each score, for the recovery experience and initial SRS-18, were calculated and entered into the regression model. If values were less than 10, then the possibility of multicollinearity was considered low [22]. Post hoc analysis was also performed following the procedure of Aiken and West [23] in the case of significant interaction effects. Prior to analysis, the study calculated two variables plugged in at 1 *SD* above or below the mean score for SRS-18 at T1, which exerted a significant effect on the improvement of SRS-18 scores. The significance level was set to *p* < 0.05.

## 3. Results

### 3.1. Demographic Characteristics

In total, 38 of the 174 participants scored zero on the SRS-18. Table 1 provides the demographic characteristics of the 38 participants and remaining 136 participants. Out of the 136 participants with SRS-18 scores of 1 or higher, 59.6%, 55.1%, and 45.6% were female, married, and parents, respectively. In addition, 76.5% and 14.7% worked at elementary and special-needs schools, respectively. The average overtime worked was 7.6 h, where 126 (92.6%) of the participants reportedly worked on weekends. Conversely, out of the 38 participants with SRS-18 scores of 0, 50.0% (*n* = 19), 65.8% (*n* = 25), and 60.5% (*n* = 19) were female, married, and parents, respectively. In addition, 84.2% (*n* = 32) and 10.5% (*n* = 4) worked in elementary and special-needs schools, respectively. The average overtime was determined to be 6.2 h, where 89.5% (*n* = 34) of the participants worked on weekends. Data for 136 participants were considered for further analysis.

### 3.2. Descriptive Statistics

Table 2 displays the means, standard deviations, and correlations between variables. The levels of psychological stress response at T2 were found to be significantly lower compared with those observed at T1 [*t*(135) = 4.34, *p* < 0.01].

### 3.3. Regression Analysis

Table 3 summarizes the results of the hierarchical regression analysis of factors predicting reduction in psychological stress response from T1 to T2. In Step 1, the control variables explained 15% of variance in the reduction of psychological stress response from T1 to T2 (*F*(7, 128) = 4.47, *p* < 0.01). The significant predictors included the presence of weekend work (β = 0.18, *p* = 0.045), level of psychological stress at T1 (β = 0.40, *p* < 0.01), and relaxation (β = 0.22, *p* = 0.046). In contrast, life stressors (β = 0.02, *p* = 0.80), psychological detachment (β = 0.05, *p* = 0.57), mastery (β = −0.09, *p* = 0.32), and control (β = 0.03, *p* = 0.77) were not identified as significant predictors.

In Step 2, the addition of four interaction terms has significantly increased the proportion of variance explained by the variables [Δ*F*(4, 124) = 3.03, Δ*R*^2^ = 0.07, *p* = 0.02], whereas the model explained 20% of variance in reduction in psychological stress response [*F*(11, 124) = 4.12, *p* < 0.01]. The interaction between psychological stress levels at T1 and relaxation has been determined to be a significant predictor of reduction in the psychological stress response (β = 0.28, *p* = 0.04). The other interaction terms were non-significant (psychological detachment: β =−0.16, *p* = 0.08; mastery, β = −0.05, *p* = 0.63; control, β = 0.09, *p* = 0.39). Analysis of the interaction indicated that the effect of relaxation on the decrease in SRS-18 scores was significant (*B* = 0.94, *SE B* = 0.31, *p* < 0.01), where SRS-18 scores at T1 were 1 *SD* higher than the mean (i.e., SRS score = 20.0); however, this effect is deemed to be non-significant (*B* = 0.01, *SE B* = 0.27, *p* = 0.96) when SRS-18 scores at T1 were 1 *SD* lower than the mean (i.e., SRS score = 1.0). Although the score was 1 *SD* lower than the average, which was 0.8, the minimum score was 1. Figure 1 illustrates these effects.

## 4. Discussion

This longitudinal study examined whether levels of the psychological stress response before the weekend moderate the relationship between increased recovery experience and reduced psychological stress response on the weekend. Data from 136 schoolteachers were analyzed. The study found that a working time of 46.4 h (7.6 h of overtime + 38.75 h of legal working time) was less than those reported in Organization for Economic Cooperation and Development [19]. In addition, the average SRS-18 scores of the present sample were 10.4 and 7.9 at T1 and T2, respectively, which were much lower than the average scores [9]. Furthermore, 38 participants were determined to be not stressed at all; therefore, this present sample represented a relatively less stressed population.

The reasons for the low levels of psychological stress response were unclear. One possibility, however, is that the teachers who were invited but declined to participate experienced higher levels of psychological stress response than the participants. For example, Bergman et al. [24] have reported that low rates of participation in a psychiatric epidemiological study were related to previous psychiatric diagnosis as well as socioeconomic variables, such as lower income, lower education, and being unmarried. Another possibly is that teachers who declined to participate observe tight schedules and experience difficulty in finding time to complete a battery of questionnaires. Future studies can formulate hypotheses building on these assumptions and test them.

The hypothesis that increases in psychological detachment, relaxation, and mastery would be associated with a reduced psychological stress response on the weekend, when the level of psychological stress before the weekend is high, was partially supported. More frequent relaxation was significantly associated with greater reduction of the psychological stress response between T1 and T2. The strength of this association depended on the degree of the psychological stress response at T1. Consistent with this hypothesis, relaxation was determined to be not associated with the extent of reduction of the psychological stress response among participants with SRS-18 scores of 1, which was the minimum score; however, this relationship was observed in participants with SRS-18 scores 1 SD higher than the mean. In contrast, the interaction between levels of psychological stress at T1 and psychological detachment or mastery did not influence the reduction of the psychological stress response.

This study was one of the first to directly examine the moderating effects of psychological stress on the relationship between recovery experience and reduction in psychological stress response from workdays to the weekend. The present results extended, and were partially consistent with those of Siltaloppi et al. [16], who conducted a cross-sectional survey and demonstrated the moderating effects of job control and time demand on this relationship; however, this study focused directly on the psychological stress response and employed a longitudinal study design, which provided a solid assessment of the moderating effects. Moreover, the results have demonstrated that the strength of the relationship between relaxation and reduction in psychological stress response from workdays to weekends is dependent on the levels of psychological stress before weekends. Moreover, results of this study added new insights to the literature on the moderating variables of the relationships between recovery experiences and the psychological stress response [14] by clarifying the psychological stress response level before the weekend.

The reason for this moderating effect remains unclear. One explanation could be that the meanings and correlates of relaxation during weekends may differ between participants with high and low levels of stress before the weekend. That is, the former could not experience relaxation, which was assessed via the REQ-J, unless stress levels were reduced. In contrast, the latter may experience relaxation despite stable levels of psychological stress. This speculation suggests that a reduction in psychological stress response is associated with relaxation during weekends in participants who were stressed before weekends; however, this relationship is not necessarily observed in those who were not stressed. Another possible explanation for this finding is the floor effect, that is, less room is provided for the improvement in psychological stress response among participants with low levels of stress, which is not the case for those with high levels of stress; however, such hypotheses should be examined empirically.

Despite the interpretation of the results being open to discussion, they provide important implications for the literature. Previous studies on recovery experience during weekends overlooked the strength of the psychological stress response before weekends. Such a lack of consideration may have distracted the researchers’ attention away from the true associations between recovery experience and changes in psychological stress response, which are difficult to identify if participants with low stress levels are included. These current results have further demonstrated the importance of considering the participants’ levels of psychological stress, and clearly illustrated that if the levels of psychological stress are not considered, then the strength of the relationship between relaxation as a recovery experience and reduction in psychological stress response from workdays to weekends cannot be examined accurately; therefore, excluding participants with little to no stress or controlling for the effects of levels of psychological stress is preferable.

Despite these important implications, this study remained subject to certain limitations. First, the moderating effect of levels of psychological stress response on the relationship between recovery experience and the reduction of psychological stress response from workdays to weekends with relatively low levels of psychological stress response was tested. Thus, the results should be interpreted with caution given this limitation. The floor effect is very likely to exist, or the psychological stress response has less space to decrease; however, these results provided one of the first pieces of evidence regarding such moderating effects. Thus, the findings indicate that testing such effects among highly stressed participants is important. Second, the sample only included 136 participants from certain areas in Japan and did not represent the entire Japanese population; however, these present findings provide a rationale for further studies, which should replicate the present findings using a larger, more diverse sample. In addition, although not experimental, the study examined the associations between recovery experience and changes in psychological stress response. Thus, further research is required to manipulate recovery experience and confirm whether such manipulation can reduce the psychological stress response. A clarification of this aspect would enhance the current understanding on recovery experience during weekends. Finally, this study found that the score of psychological stress response decreased by 2.5 points from T1 to T2. Although the magnitude of the reduction is significant, it is unclear whether it is meaningful for individuals. It is important to examine this point.

In conclusion, this preliminary study found that the strength of the relationship between relaxation and reduction in the psychological stress response from workdays to weekends is dependent on the levels of psychological stress before weekends. The strength of this relationship was stronger in participants with high levels of psychological stress before weekends than those with low levels. In addition, these findings demonstrated the importance of considering the level of psychological stress response during weekdays when researching the recovery experience during weekends.

## Figures and Tables

**Figure 1 behavsci-12-00163-f001:**
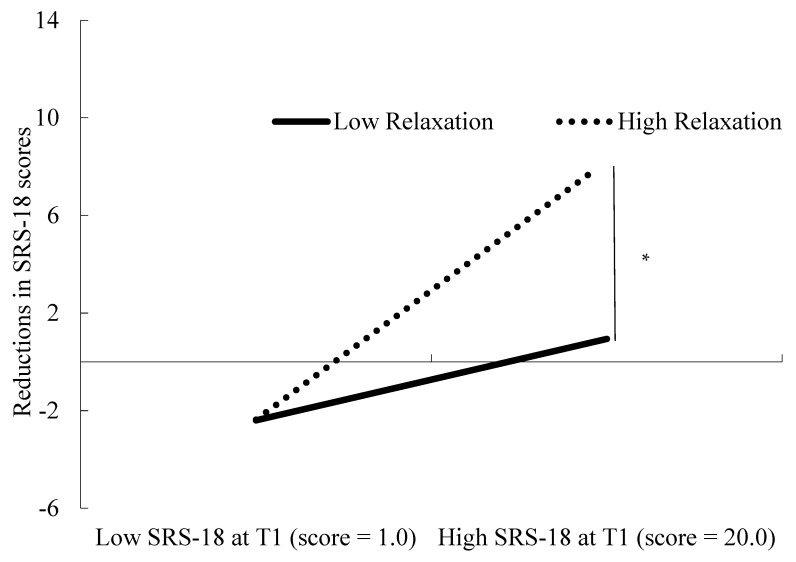
Effect of the significant interaction between the psychological stress response at T1 and relaxation on the reduction of the psychological stress response. T1: Time 1; SRS-18: Stress Response Scale-18. *****
*p* < 0.01.

**Table 1 behavsci-12-00163-t001:** Demographic characteristics of participants included in (*n* = 136) and excluded from (*n* = 38) analysis.

	Included Participants		Excluded Participants	
	*n* (%)	*M* (*SD*)	*n* (%)	*M* (*SD*)
Age (years)		40.7 (10.51)		41.8 (9.91)
Women	81 (59.6)		19 (50.0)	
Married	75 (55.1)		25 (65.8)	
Having a child or children	62 (45.6)		23 (60.5)	
School at which participant works				
Elementary	104 (76.5)		32 (84.2)	
Junior high school	11 (8.1)		1 (2.6)	
High school	1 (0.7)		1 (2.6)	
Special-needs school	20 (14.7)		4 (10.5)	
Managerial position	4 (2.9)		0	
Type of teacher				
Teacher	116 (85.3)		35 (92.1)	
Nursing teacher	20 (14.7)		3 (7.9)	
Caregiving of family member(s)	10 (7.4)		1 (2.6)	
Weekend work	126 (92.6)		34 (89.5)	
Average weekly overtime (hours)		7.6 (7.56)		6.2 (5.69)

**Table 2 behavsci-12-00163-t002:** Means, standard deviations, and correlations between variables (*n* = 136).

Variables	Mean (*SD*)	1	2	3	4	5	6	7
1. Psychological stress response (T1)	10.4 (9.60)							
2. Psychological stress response (T2)	7.9 (9.50)	0.75 **						
3. Reduction in psychological stress response	2.5 (6.80)	0.37 **	−0.34 **					
4. Life stressors	9.2 (2.56)	0.25 **	0.20 *	0.07				
5. Psychological detachment	8.7 (3.35)	−0.14	−0.15	0.01	−0.15			
6. Relaxation	14.5 (3.86)	−0.34 **	−0.37 **	0.03	−0.37 **	0.43 **		
7. Mastery	8.6 (2.99)	−0.15	−0.12	−0.05	−0.08	0.18 *	0.33 **	
8. Control	12.2 (3.00)	−0.15	−0.18 *	−0.04	−0.31 **	0.13	0.48 **	0.38 **

T1: Time 1 (Friday); T2: Time 2 (Sunday). * *p* < 0.05. ** *p* < 0.01.

**Table 3 behavsci-12-00163-t003:** Results of the hierarchal regression analysis of predictors of the reduction of psychological stress response from Time 1 to Time 2 (*n* = 136).

	Step 1		Step 2	
	β	*t*	β	*t*
Life stressors	0.02	0.25	0.02	0.25
Weekend work	0.18	2.02 *	0.17	1.96
Psychological stress response at T1	0.40	4.60 **	0.51	5.55 **
Psychological detachment	0.05	0.57	0.03	0.30
Relaxation	0.22	2.01 *	0.27	2.51 *
Mastery	−0.09	−1.01	−0.10	−1.17
Control	0.03	0.29	−0.01	−0.07
Psychological stress response at T1 × Psychological detachment			−0.16	−1.75
Psychological stress response at T1 × Relaxation			0.28	2.09 *
Psychological stress response at T1 × Mastery			−0.05	−0.48
Psychological stress response at T1 × Control			0.09	0.86
*R^2^* (adjusted *R^2^*)	0.20 (0.15)		0.27 (0.20)	
*F*	4.47		4.12	
*ΔR^2^*	0.20		0.07	
*F*	4.47		3.03	

T1: Time 1. * *p* < 0.05. ** *p* < 0.01.

## Data Availability

The dataset of this study is available from the corresponding author on reasonable request.

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
