# Peer review of "Unwinding on the Weekend from Work-Related Stress: Moderating Effect of Weekday Psychological Stress on the Relationship between Increased Recovery Experience and Reduction of Psychological Stress on the Weekend"

_behavsci, 2022, doi:10.3390/bs12060163_

Round 1

Reviewer 1 Report

This manuscript represents what I believe to be a well-conducted study in many ways. This is particularly true of the statistical approach using hierarchical regression, with relevant control variables included, and a well-defined strategy for reducing multicollinearity. The tables and figure are straightforward for readers to interpret and appropriate.

My main concern is that the title and introduction are confusing. First, I would suggest trying to edit the title so that it is shorter, clear, and likely to attract readers. Perhaps something beginning with: Unwinding at the weekend from work-related stress…

I struggled to understand the use of the term ‘moderating variable’ in the context of this research. The problem is that a moderating variable in science is on the causal path between two variables, but this is not how the term is meant in this manuscript, which is that relaxation and other behaviours at the weekend will alleviate stress, and this effect will be greater for workers who were more stressed during the week.

I question whether the research is exploratory: usually exploratory research does not have a well-defined hypothesis, and this research has a formal hypothesis.

More detail is required on the recovery experience variables (REQ-J). Understanding what exactly these consist of is critical for understanding the analysis. What are the validity and reliability stats for the REQ-J?

Explain the expected direction of the interaction effects. At first I expected stress at T1 x Relaxation to have a negative association with stress at T2 – but this is not the case, as can clearly be seen in the interaction plot.

In the discussion please explain the magnitude of the reduction in reported stress between T1 and T2 using words: what does a difference of 2.5 in stress scores mean for an individual? Is it really a big difference?

From a philosophical scientific perspective, I am not sure about removing the observations with stress scores of 0 at T1. Teachers with very low stress scores at T1 also did not have the possibility of recovery because they weren’t stressed in the first place. Surely the interaction effect should logically be present for this reason, because an interaction is highly likely in the first place. The important variables are not the interaction terms but the main effects, which should be stronger with the interaction terms present simply because a person who is less stressed at T1 can’t experience a lot of recovery. Given this, the fact that the main effect for relaxation was larger with the interaction effect included in the model is the strongest piece of evidence in the study.

Author Response

Dear Reviewer 1

Thank you very much for reviewing our submission and providing us with constructive feedback. We have revised the manuscript based on the comments from you and the other reviewer. Our responses to review comments are in bold and are written after the arrow (à). In addition to the changes described in this letter, we have made minor edits. In the MAIN document, the sentences that have been added are highlighted in red.

My main concern is that the title and introduction are confusing. First, I would suggest trying to edit the title so that it is shorter, clear, and likely to attract readers. Perhaps something beginning with: Unwinding at the weekend from work-related stress…

- The title was modified as follows. We believe that the new title is more clear and more attractive.

(In the original manuscript)

Weekday psychological stress response as a moderating variable in the relationship between increased recovery experience and reduction in psychological stress response during weekends: Recovery experience during weekends

(In the revised manuscript)

Unwinding on the weekend from work-related stress: Moderating effect of weekday psychological stress on the relationship between increased recovery experience and reduction of psychological stress on the weekend

I struggled to understand the use of the term ‘moderating variable’ in the context of this research. The problem is that a moderating variable in science is on the causal path between two variables, but this is not how the term is meant in this manuscript, which is that relaxation and other behaviours at the weekend will alleviate stress, and this effect will be greater for workers who were more stressed during the week.

- Thank you very much for your feedback on important points. Indeed, as you pointed out, a moderating variable in science is on the causal path between two variables. However, previous cross-sectional studies we cited in this manuscript and other studies in occupational health psychology have used the term “moderating variable” (probably for convenience). Therefore, we continue to use the term “moderating variable” and “moderate”.

I question whether the research is exploratory: usually exploratory research does not have a well-defined hypothesis, and this research has a formal hypothesis.

- Thank you very much for your feedback on important points. We have used the term “preliminary” instead of “exploratory” as follows.

(Line 17, Line 374)

More detail is required on the recovery experience variables (REQ-J). Understanding what exactly these consist of is critical for understanding the analysis. What are the validity and reliability stats for the REQ-J?

- Examples of items of the (REQ-J) and information about reliability of the REQ-J in the present sample have been added as follows to the Measures section (page 4).

An example of an item from psychological detachment is “I forget about work.” One from relaxation is “I use the time to relax.” One from control is “I learn new things.” One from mastery is “I decide my own schedule.”

 (Line 162-164)

  Cronbach’s alpha coefficients for the present sample were .81, .87, .88., and .85 for psychological detachment, relaxation, control, and mastery, respectively.

(Line 168-170)

In addition, the following sentence was modified to clarify what had been measured with this questionnaire.

(In the original manuscript)

Items were then rated using a 5-point Likert-type scale ranging from 1 (strongly disagree) to 5 (strongly agree).

(In the revised manuscript)

Items were then rated using a 5-point Likert-type scale ranging from 1 (strongly disagree) to 5 (strongly agree) to best represent what each respondent did during the weekend.

(Line 166)

Explain the expected direction of the interaction effects. At first I expected stress at T1 x Relaxation to have a negative association with stress at T2 – but this is not the case, as can clearly be seen in the interaction plot.

- The dependent variable was not a T2 score but was the reduction of the score of psychological stress response. The sentences in the method and results sections as well as the title of Table 3 were misleading. I am very sorry. To clearly indicate what the dependent variable was, the following sentences have been added.

The dependent variable was the change in the psychological stress response from Time 1 to Time 2, which was calculated by subtracting the Time 1 score from the Time 2 score. Greater values represent greater stress reduction.

(Line 207-209, Line 252-254)

  The title of Table 3 has been revised as follows:

(In the original manuscript)

Results of hierarchal regression analysis of predictors of psychological stress response at Time 2 (n = 136).

(In the revised manuscript)

Results of the hierarchal regression analysis of predictors of the reduction of

psychological stress response from Time 1 to Time 2 (n = 136).

(Page 6)

In the discussion please explain the magnitude of the reduction in reported stress between T1 and T2 using words: what does a difference of 2.5 in stress scores mean for an individual? Is it really a big difference?

- Thank you very much for your feedback on important points. It is unclear whether a decrease of 2.5 in the stress score is meaningful or not. We have noted this as a limitation as follows:

Finally, this study found that the score of psychological stress response decreased by 2.5 point from T1 to T2. Although the magnitude of the reduction is significant, it is unclear whether it is meaningful for individual. It is important to examine this point.

(Line 370-373)

From a philosophical scientific perspective, I am not sure about removing the observations with stress scores of 0 at T1. Teachers with very low stress scores at T1 also did not have the possibility of recovery because they weren’t stressed in the first place. Surely the interaction effect should logically be present for this reason, because an interaction is highly likely in the first place. The important variables are not the interaction terms but the main effects, which should be stronger with the interaction terms present simply because a person who is less stressed at T1 can’t experience a lot of recovery. Given this, the fact that the main effect for relaxation was larger with the interaction effect included in the model is the strongest piece of evidence in the study.

- Thank you very much for your feedback on important points. We have mentioned the significant main effect of relaxation on the reduction of psychological stress response as follows:

More frequent relaxation was significantly associated with greater reduction of the psychological stress response between T1 and T2. The strength of this association de-pended on the degree of psychological stress response at T1.

(Line 306-308)

Reviewer 2 Report

overall, well-done and well-written.

minor recommendation - while the authors did well in showing the value and concerns about the results for the scholarly literature, I don't see much application to the practitioner-literature. Adding a paragraph to the discussion section about the value of your findings for the practitioner would benefit the reader.

very minor recommendation. throughout the document there are vague beginnings to sentences such as page 1 lines 29-30 "Previous research has suggested that weekends can serve as an opportunity for recovery 29
from work-related stress." Clarity can be improved if the writing simple began with the citations of the reverenced sources so that it is clear the authors are not alluding to 'all' prior research but to specific sources. This very minor concern continues throuhout.

Author Response

Dear Reviewer 2

Thank you very much for reviewing our submission and providing us with constructive feedback. We have revised the manuscript based on the comments from you and the other reviewer. Our responses to review comments are in bold and are written after the arrow (à). In addition to the changes described in this letter, we have made minor edits. In the MAIN document, the sentences that have been added are highlighted in red.

minor recommendation - while the authors did well in showing the value and concerns about the results for the scholarly literature, I don't see much application to the practitioner-literature. Adding a paragraph to the discussion section about the value of your findings for the practitioner would benefit the reader.

- I agree with the opinion that adding a paragraph to the discussion section about the value of your findings for the practitioner would benefit the reader. However, this study is preliminary and is for the scholarly literature. It is impossible to mention any practical implications at this stage.

very minor recommendation. throughout the document there are vague beginnings to sentences such as page 1 lines 29-30 "Previous research has suggested that weekends can serve as an opportunity for recovery 29from work-related stress." Clarity can be improved if the writing simple began with the citations of the reverenced sources so that it is clear the authors are not alluding to 'all' prior research but to specific sources. This very minor concern continues throuhout.

- These issues have been addressed after the manuscript has been formatted.

Round 2

Reviewer 2 Report

the authors replied to my comments. I can accept the author's position that this is a scholarly study and that application is not required. I encourage the authors to always point applied researchers toward why applied research might be appropriate give what you found in your scholarly study.

Author Response

Thank you very much for your comment.